# SemSA: Semantic Sparse Attention is hidden in Large Language Models.

## Abstract

Sparse attention is one of the most effective approaches for addressing the $O(N^2)$ attention complexity of transformer models. Existing methods manually designs a uniform sparse attention mask for all attention heads. However, uniform masks treat different attention heads equally. To preserve necessary attentions for important heads, the masks are unnecessarily dense for unimportant heads, limiting the overall sparsity and wall-clock speedup. Thus, we propose Semantic Sparse Attention (SemSA) paradigm. It uses statistical information to evaluate, generate and optimize different sparse attention masks for different heads. We observe that the acquired attention masks successfully learn different semantic information from the dense pre-trained large language models: some heads focus on contents while others mainly encode the token positions. We optimize SemSA GPU operators and evaluate it on popular large language models OPT-6.7B (2k tokens) and Llama2-7B (4k tokens). Compared with dense PyTorch models, SemSA achieves $4.18 \sim 11.67\times$ and $1.36 \sim 2.34\times$ speedup for attention layer and first-token-latency with negligible accuracy loss. Compared with other sparse attention methods optimized with state-of-the-art sparse framework, SemSA achieves up to $1.6\times$ sparsity, $1.4\times$ attention speedup with higher accuracy.

## 1 Introduction

Transformer is an increasingly popular model architecture in a wide range of applications (Brown et al., 2020; Dosovitskiy et al., 2020), including natural language processing, computer vision, and so on. It has played a vital role in the remarkable achievements of recent large language models (LLMs). At the heart of Transformer models lies the attention layer that takes a sequence of $N$ input tokens, and computes the correlation between them with $O(N^2)$ complexity. With the growing scale of tokens, the squared complexity of the attention layer makes it a key latency bottleneck. For example, for the large language model Llama2-7B (Touvron et al., 2023) with a context length of 4K, the attention layers account for over 70% latency.

Previous work designs approximate methods to reduce the long latency of attention, which are widely used for MB-scale models like BERT (Devlin et al., 2018). Linear attention approaches avoid the explicit computation of the $O(N^2)$ attention matrix and re-trains the model. They project the token dimension $N$ to a fixed hidden dimension (Wang et al., 2020), or multiply key and value matrix before multiplying query matrix (Qin et al., 2022). These approaches require an extreme re-training overhead for LLMs, as linear attention methods need different weights and hidden states from existing dense attention models because of their fundamentally different structure. In contrast, sparse attention approaches (Zaheer et al., 2020; Beltagy et al., 2020; Child et al., 2019; Feng et al., 2022) would require much fewer training cost. These approaches design masks to skip the computation of some attention values, with the proportion of the skipped positions defined as the *sparsity*. These existing sparse attention methods use uniform masks, meaning that each attention head has the same sparsity and the same type of mask pattern. This uniformity ignores the semantic differences among attention heads and limits the sparsity due to the presence of sensitive heads.

In this work, to push the limit of the sparsity and thereby unleash the latency benefit of sparse attention methods, we propose SemSA. The main idea of SemSA is to leverage the semantic knowledge learned by dense LLMs to decide the sparse masks that retain essential attention patterns. Compared with previous masks that manually define uniform attention mask pattern or density, SemSA

preserves different semantics for different heads, yielding lower density. Our contributions are summarized as follows.

- We propose the semantic sparse attention method. It reduces the mask density by automatically optimizing the sparse pattern for each attention head with information from the dense model. Both intra-head and inter-head impacts of the sparse mask are modeled to minimize accuracy impact.
- We invest the generated masks and show their interpretable semantics. We find that our generated masks successfully captures the token-position-based and token-content-based attention pattern of different attention heads.
- We design GPU kernels to support SemSA's sparse attention computation. Our implementation balances the workloads of different GPU kernels and achieves $1.74\times$ attention speedup (for one attention layer in Llama2-7B with a 4K context) over state-of-the-art sparse attention acceleration framework xFormers.

We compare SemSA with other sparse attention methods on popular large language models OPT-6.7b and Llama2-7B. SemSA demonstrates $4.18 \sim 11.67\times$ attention wall-time speedup over dense PyTorch models, while maintaining high accuracy. Compared to other sparse attention methods, SemSA achieves $1.27\times$ to $1.6\times$ mask density reduction and while achieving better accuracy. Our code is available at `https://anonymous.4open.science/r/SemSA/README.md`.

## 2 RELATED WORK

### 2.1 ATTENTION MECHANISM

Attention generally refers to Multi-Head self Attention (MHA) mechanism (Vaswani et al., 2017). Given $B$-batched input sequence of length $N$, a linear projection transform the token embeddings into three matrices: query matrix $\mathbf{Q}$, key matrix $\mathbf{K}$, and value matrix $\mathbf{V}$ of size $\mathbb{R}^{(B \times H) \times N \times d}$. $H$ and $d$ denote the number of attention heads and the hidden dimension of each head.

As shown in equation 1, the attention matrix $A$ is the multiplication of $\mathbf{Q}$ and $\mathbf{K}^T$ with the softmax activation function. For generative language models (Brown et al., 2020; Zhang et al.; Touvron et al., 2023), a casual mask $M$ is applied to mask out the upper triangular elements of the attention matrix. It is commonly implemented by adding negative infinite to the upper triangular elements of $A$ before softmax. Finally, attention output $O$ is the the multiplication of $\mathbf{O}$ and $\mathbf{V}$.

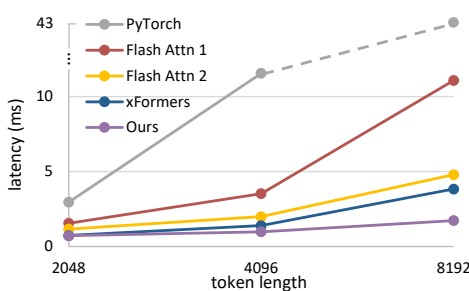

Figure 1: Attention latency of the Llama2-7B model with different input token length and attention framework.

$$\mathbf{S} = \mathbf{Q}\mathbf{K}^T \in \mathbb{R}^{(B \times H) \times N \times N}, \mathbf{A} = \text{softmax}(\mathbf{S} + \mathbf{M}) \in \mathbb{R}^{(B \times H) \times N \times N}, \mathbf{O} = \mathbf{A}\mathbf{V} \in \mathbb{R}^{(B \times H) \times N \times d}$$
(1)

### 2.2 EFFICIENT ATTENTION

Efficient attention methods are proposed to avoid the costy attention matrix computation in equation 1. A branch of work uses linear methods to approximate the attention computation. Wang et al. (2020); Choromanski et al. (2020) approximate the query and value matrices by projecting their $N$ dimension onto a lower $D$-dimension. Choromanski et al. (2020); Qin et al. (2022); Peng et al. (2022) approximate the softmax activation with a kernel function $\phi$ to supplement $\text{softmax}(QK)^T V$ with $\phi(Q)(\phi(K)^T V)$. It avoids generating the large $N \times N$ attention matrix. KDE approximation (Zandieh et al., 2023) and recursive attention computation (Poli et al., 2023; Feng et al., 2022) methods are also proposed. The benefit of linear complexity and regular computation makes them

Figure 2: SemSA pipeline to generate attention mask. Given a trained LLM, SemSA first evaluates the effect $E$ of each attention matrix $A$ to the loss $L$. Then it generates a set of candidate masks for each attention head and evaluates their quality score $s$. Finally, it solves an optimization problem to select the masks that preserve the most important attentions under the average density bound $d_t$.

popular for the BERT (Devlin et al., 2018) small encoder model family. However, the new computation scheme needs different weights from the vanilla transformers, posing much re-training overhead for large language models. Besides, some assumptions, like the low rank assumption of the attention matrices (Wang et al., 2020), do not hold for generative models (Dong et al., 2021).

The other branch of work use sparse masks to substitute the casual mask $M$ to skip the computation of some attention values. Dynamical sparse attention (Kitaev et al., 2020; Roy et al., 2021; Wang et al., 2021; Lu et al., 2021; Qu et al., 2022) uses low-cost operations to decide the mask for each sentence. Forexample, (Kitaev et al., 2020; Roy et al., 2021) use hashing to chunk the sentences and mask out attentions between different chunks. However, due to the complex control and computation flow, dynamic sparse attention methods often require specific hardware to achieve wall-time speedup (Wang et al., 2021; Lu et al., 2021; Qu et al., 2022). Static sparse attention (Zaheer et al., 2020; Beltagy et al., 2020; Child et al., 2019; Dai et al., 2023) manually pre-defines masks and directly use them for all sentences. Thanks to the fixed computation flow, static sparse attention is generally more efficient and GPU-friendly. However, manually designed masks have no awareness of the sentence data distribution, which limit their performance with low mask density. Trivial GPU kernel implementations also show limited acceleration due to load imbalance and random attention data access.

### 2.3 ATTENTION ACCELERATION FRAMEWORK

Previous works also propose acceleration frameworks using system optimizations. Flexgen (Sheng et al., 2023) minimizes GPU memory requirements with active memory swap. Flash attention 1 (Dao et al., 2022) and 2 (Dao, 2023) use different dataflow optimizations to reduce memory footprint for attention speedup. xFormer (Lefaudeux et al., 2022) uses block sparse GPU kernels to regulate the memory and computation of sparse attention.

## 3 SEMANTIC SPARSE ATTENTION (SEMSA)

In this section, we introduce each step of the semantic sparse mask generation pipeline as shown in Figure 2. SemSA first **evaluates** the effect $E$ of each attention value to the model's final prediction. Given the density $d$ for an attention head, SemSA **generates** the corresponding candidate mask based on the effect. Then, SemSA calculates the quality score $s$ of each mask $M$. Finally, after calculating the quality scores for the mask corresponding to each density choice in each attention head, SemSA **optimizes** the overall quality under a constraint of the overall density to decide the suitable density for each head.

### 3.1 ATTENTION EFFECT EVALUATION

Given a trained LLM with dense attention, SemSA first analyzes each attention value and its gradient to identify the unimportant attention positions to mask out. We assume that the change of the model prediction loss $L$, caused by attention mask, can be approximated as the first-order Taylor expansion of the change of attention matrices $A$, as shown in equation 2.

$$\Delta L = \sum_h \sum_i \sum_j \frac{\partial L}{\partial A_{h,i,j}} \cdot \Delta A_{h,i,j}, \tag{2}$$

where $h \in [0, \#\text{layer} \times \#\text{head per layer})$ denotes the attention head index, and $i, j$ denote the row and column index of the attention matrix.

Because the softmax function normalizes the summation of each row of attention matrix to one, masking out one attention value $A_{h,i,j}$ to zero will increase other attention values in the same row $A_{h,i,n}, n \neq j$. We can simply model such intra-row influence on $\Delta \boldsymbol{A}$ with Lemma 3.1.

**Lemma 3.1.** *When masking out attention value $A_{h,i,j}$ at head $h$, row $i$, and column $j$, it also influences the attention values in the same row by $\Delta A_{h,i,n|j}$.*

$$A_{h,i,n} = \frac{e^{S_{h,i,n}}}{\sum_j e^{S_{h,i,j}}}$$

$$\Delta A_{h,i,n|j} = \begin{cases} -A_{h,i,n}, & n = i \\ A_{h,i,n}(\sum_j e^{S_{h,i,j}} / \sum_{j \neq n} e^{S_{h,i,j}} - 1), & n \neq j \end{cases} \tag{3}$$

We define the effect matrix $\boldsymbol{E}_h$ to quantify how much influence will masking each attention value cause to the final prediction loss based on equation 2. Note that the influence considers the attention value variance of the value itself and the values in the same row.

$$E_{h,i,j} = \left| \sum_n \frac{\partial L}{\partial A_{h,i,n}} \cdot \Delta A_{h,i,n|j} \right| \tag{4}$$

Given equation 4 and Lemma 3.1, we derive the following theorem.

**Theorem 1.** *Given attention matrix $\boldsymbol{A}_h \in \mathbb{R}^{N \times N}$ of head $h$ and its gradient $\partial L / \partial \boldsymbol{A}_h \in \mathbb{R}^{N \times N}$, the effect $E_h \in \mathbb{R}^{N \times N}$ of masking each attention value is defined as follows.*

$$\boldsymbol{E}_h = \left| \frac{\boldsymbol{A}_h}{1 - \boldsymbol{A}_h} \cdot \left( \frac{\partial L}{\partial \boldsymbol{A}_h} - (\frac{\partial L}{\partial \boldsymbol{A}_h} \cdot \boldsymbol{A}_h) \mathbb{1}^{N \times N} \right) \right| \tag{5}$$

With theorem 1, we design the **evaluation** step to quantify the attention values' impact on the final prediction. SemSA performs the standard next-token-prediction task on a small dataset to calculate the cross entropy loss $L$. The effect is averaged over different sentences. The average attention effect $\bar{E}$ guides how SemSA generates the masks to only skip the attention values with the minimum average impact on the final prediction.

## 3.2 ATTENTION MASK GENERATION

Given a density, SemSA **generates** candidate sparse masks based on the attention effect for each attention head. We follow previous work (Zaheer et al., 2020; Child et al., 2019) to use the block sparse matrix, so as to avoid random memory access and maximize wall-time speedup. For each block of size $b$, all the attention values are either all masked or not masked. We use the average attention effect of each block to quantify the importance of the block.

For notion simplicity, we define the layout $L \in \mathbb{R}^{H \times N/b \times N/b}$ matrix to represent the effect of each block of the attention matrix. Note that generative LLMs only calculate the attention value as lower triangular matrix (Brown et al., 2020; Zhang et al.; Touvron et al., 2023), Thus their diagonal blocks have fewer elements than others.

$$L_{h,i,j} = \sum_{m \in [bi, b(i+1))} \sum_{n \in [bj, b(j+1)), n \leq m} \bar{E}_{h,m,n} / \sum_{m \in [bi, b(i+1))} \sum_{n \in [bj, b(j+1)), n \leq m} 1 \tag{6}$$

SemSA preserves $k$ blocks in each row that have the largest average attention effect and masks out the others. Note that SemSA can generate different candidate masks with different $k$, which controls the density of the masks. In practice, SemSA generates a set of candidate masks for each head with a given set of density $\{d_i\}$.

Figure 3: Semantic analysis on Llama2. (a) SemSA's mask density of different heads and layers. (b) A typical position-dominated head, where each token always attends to its previous token. SemSA aggressively masks out attention values at other positions. (c) A typical token-dominated head, where different input tokens results in different attention pattern. SemSA preserves most attention values since important attention positions are hard to pre-determine.

$$T_{h,i,j}^{(d)} = \begin{cases} 0, & L_{h,i,j} \in \text{Top}_k \\ -\inf, & L_{h,i,j} \notin \text{Top}_k \end{cases}, \text{Top}_k = \{l_{h,i,j_r}, r < k, k = dN/b | l_{h,i,j_0} > l_{h,i,j_1} > \ldots \}$$

$$M_{h,i,j}^{(d)} = T_{h,\lfloor i/b \rfloor, \lfloor j/b \rfloor}^{(d)}$$

(7)

To achieve the balance between efficiency and prediction quality, SemSA evaluates the quality of each candidate masks to perform overall optimization in the next step. Each mask is given a quality score $s$ to evaluate how much variation it will cause to the loss of the final prediction. Recall that the mean effect matrix $\bar{E}$ already provides such information for each attention value, SemSA defines the scoring function $\mathbb{S}$ to get the score $s_h^{(d)}$ of the density $d$ for the attention head $h$ in equation 8.

$$s_h^{(d)} = \mathbb{S}(\boldsymbol{M}_h^{(d)} | \boldsymbol{E}_h) \quad = \left( \mathbb{1}^{1 \times s} (\boldsymbol{M}_h^{(d)} \cdot \boldsymbol{E}_h) \mathbb{1}^{s \times 1} \right) / \left( \mathbb{1}^{1 \times s} \boldsymbol{E}_h \mathbb{1}^{s \times 1} \right)$$

(8)

After the **generation** step, SemSA assigns each attention head a set of candidate masks with different densities, along with the respective mask quality score. SemSA solves an optimization problem to select the mask for each head to achieve both low mask density and high prediction quality.

## 3.3 ATTENTION DENSITY OPTIMIZATION

With the generated mask and its score, SemSA optimizes the overall mask quality given the overall density bound $d_t$. It is done by selecting the density for each head to maximize the summation of mask quality score under the constraint for the overall density as shown in equation 9.

$$\max S = \sum_h s_h, s_h \in \{s_h^{(d_h)} | d_h \in \{d_0, d_1, \ldots, d_s\}\} \quad \text{s.t.} \quad \frac{1}{H} \sum_h d_h \leq d_t$$

(9)

Intuitively, some attention heads are more concentrated, meaning that most attention values are small while few values are very large. Such heads only need a small density to preserve most of the important attentions. In contrast, some attention heads are more dispersive with few small attention values. Such heads require more density to preserve most attention values. Note that the density $d_h$ for candidate masks is selected from a pre-defined set, so we use mixed integer planning to formulate and solve the optimization problem.

## 4 SEMANTIC ANALYSIS

In this section, we invest the masks acquired with SemSA and show the interpretable semantics of the masks. Previous works manually restrict the attention pattern of the model, which may harm the semantics learned by the dense model. In contrast, SemSA preserves the semantics with statistic analysis and optimization. We use visualization, human interpretation and quantitive methods to analyze the semantics of the original model and to verify whether SemSA captures such semantics.

## 4.1 MASK VISUALIZATION AND SEMANTIC CATEGORIZATION

Given any token, two kinds of information are used as the model inputs: position encoding and token embedding. Position encoding indicates the absolute (Zhang et al.) or relative positions (Touvron

et al., 2023) of tokens in the sentence. Token embedding maps different tokens as different vectors. The attention head $h$ responds to both information and output the corresponding attention value $A_h$. As shown in equation 10, we denote the influence of position and token of head $h$ as function $P_h$ and $T_h$, respectively. The attention value $A_{h,i,j}$ between the $i$th and $j$th token $t_i$ and $t_j$ is determined by the combination $f_h$ of position and token influence functions.

$$A_{h,i,j} = \mathbb{A}_h(t_i, t_j, i, j) = f_h\left(P_h(i, j), T_h(t_i, t_j)\right) \tag{10}$$

Figure 3 visualizes two typical heads that are either dominated by position $P$ or token $T$ function. It shows the largest attention values between tokens of the example sentence, as well as the attention matrix averaged over 128 different sentences on the RedPajama dataset (Computer, 2023).

For the attention head in Figure 3(b), the local and global positional attention is clearly observed. In this head, whatever sentences are given, each token pays major attention to the first token and the prior token. As a result, the mean attention matrix accumulates extremely large attention values at the first column and the sub-diagonal. In contrast, the attention head in Figure 3(c) lays more emphasis on content-based attention. In this example, most tokens pay attention on definite article *the*, pronoun *their* and conjunction *and*. These words play important rule in the sematic structure of the sentence. Since the position distribution of important tokens are generally random, the attention matrix can show large attention values at any position. It results in a mean attention matrix without extreme mean attention values.

In conclusion, the mean attention matrix of different sentences provides a valuable insight of whether attention values of an attention head is more position-based or content-based. Intuitively, the more uneven the attention matrix value distribution is, the more position-based the head is.

## 4.2 QUANTITATIVE SEMANTIC ANALYSIS

We quantify how much the attention head is position-based and analyze whether SemSA successfully utilizes such semantics through the evaluate-generate-optimization pipeline. We model equation 10 with a linear approximation. $P_h$ and $T_h$ are random variables with the same expectation $\mu$ and standard variance $\delta$ for all heads. For attention head $h$, the weight factor $\alpha_h$ evaluates the relatively influence of position and token to the final attention value.

$$A_{h,i,j} = \alpha_h P_h(i, j) + (1 - \alpha_h)T_h(t_i, t_j) \tag{11}$$

Given the randomness of token positions in long context, we assume that the token position and its content are irrelevant. For different sentences $s$, the expectation $\mathbb{E}_t$ of the attention value between position $i$ and $j$ can be expressed as follows. Note that it excludes the matrix diagonal since $T_h(t_i, t_j), i \neq j$ and $T_h(t_i, t_i)$ may follow different distributions.

$$
\begin{aligned}
\mathbb{E}_t[A_{h,i,j}] &= \frac{1}{S} \sum_{s=1}^{S} \left( \alpha_h P_h(i,j) + (1-\alpha_h)T_h(t_i^{(s)}, t_j^{(s)}) \right) \\
&= \alpha_h P_h(i,j) + (1-\alpha_h)\frac{1}{S}\sum_{s=1}^{S} T_h(t_i^{(s)}, t_j^{(s)}) \\
&= \alpha_h P_h(i,j) + (1-\alpha_h)\mu_T, \forall i > j
\end{aligned}
\tag{12}
$$

The standard division $\sigma_p$ of $\mathbb{E}_t$ over different positions of the attention matrix is

$$
\begin{aligned}
\sigma_p(\mathbb{E}_t[A_{h,i,j}]) &= \sqrt{\frac{2}{(1+N)N} \sum_{i,j\in[1,N),i>j} [(\alpha_h P_h(i,j) + (1-\alpha_h)\mu_T) - (\alpha_h \mu_P + (1-\alpha_h)\mu_T)]^2} \\
&= \alpha_h \delta_p
\end{aligned}
\tag{13}
$$

We name $\sigma_p(\mathbb{E}_t[A_{h,i,j}])$ the Standard division of Expectation (SoE) of head $h$. Note that the expectation is taken over different sentences, while the standard division is taken over different attention positions. Since $\delta_p$ is the same for all heads, we derive that the position impact $\alpha_h$ is proportional to the SoE of different heads.

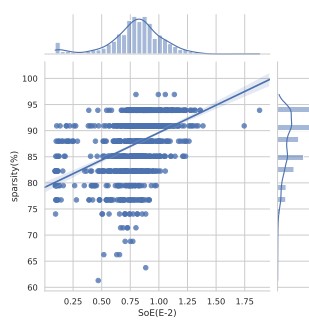

Figure 4: Positive correlation between SemSA's mask sparsity and head's dependency on position (SoE).

The conclusion quantifies the observation stated in Section 4.1. Intuitively, SoE shows how uneven the mean attention matrix is, thus showing the influence of position to the attention values. SemSA's generated mask density shows positive relation with SoE, suggesting that SemSA successfully captures the semantic information of the dense language model as shown in Figure 4.

## 5 Sparse Kernel Design

We drew inspiration from the computational framework of Flash Attention (Dao et al., 2022) and develop our own sparse kernel. In the Flash Attention algorithm, the inputs $\mathbf{Q}, \mathbf{K}, \mathbf{V}$ are partitioned into blocks. Each block of $\mathbf{Q}$ is then multiplied with the corresponding blocks of $\mathbf{K}$. The output of each block is scaled and aggregated to yield the correct result. This design allows for parallelization of computations, facilitating efficient acceleration on GPU architectures.

To skip unnecessary data loading and computation of zero-blocks, we introduced a Look Up Table (LUT). In the aforementioned process, only the non-masked blocks of $\mathbf{K}$ need to be loaded. The LUT serves as a guide, indicating which blocks of $\mathbf{K}$ should be loaded into SRAM for multiplication with $\mathbf{Q}$.

SemSA also optimizes workload load balance of the kernel. For each head, the LUT has the same shape $[N, n]$, where $N$ is the number of blocks of $\mathbf{Q}$, while $n$ is the number of blocks of $\mathbf{K}$ that will be loaded into SRAM. For each row, the number of blocks to be loaded and computed is the same, which helps to balance the GPU workload as mentioned in equation 7.

We implement our sparse kernel based on Triton (Tillet et al., 2019) and Flash Attention. In Flash Attention, inputs $\mathbf{Q}, \mathbf{K}, \mathbf{V}$ are split into blocks, and the non-masked blocks are computed. Since not all the blocks contribute the same, SemSA's evaluate-generate-optimization pipeline masks out most blocks. So we design our sparse kernel, using $\mathbf{lut} \in \mathbb{R}^{H \times numblocks \times nnz}$ to indicate which block in $\mathbf{K}$ will be loaded into SRAM. The less block is computed, the more acceleration we can achieve.

## 6 Experiment

We evaluate the effectiveness of SemSA from the perspective of both accuracy and efficiency. The comprehensive evaluation is done via perplexity analysis, as well as long sequence generation tasks. We also analyze the validity of each component of SemSA through extensive ablation studies.

### 6.1 Experiment Setup

In this section, we describe the experimental setup and results employed to evaluate the performance of our approach. SemSA's attention effect evaluation is performed on a small sampled subset of Red-Pajama dataset Computer (2023) with 128 rows (3.5M tokens). For finetuning, all the models are finetuned with another sampled RedPajama dataset consisting of 8192 rows (56M tokens) We run the perplexity evaluation on error-sensitive (Yao et al., 2022) WikiText-103 and WikiText2 (Merity et al., 2016) dataset. We also use 3 datasets from LongBench (Bai et al., 2023) for long-context text generation tasks, covering a wide range of code completion(lcc), single-document QA(multifieldqa en) and few-shot learning(samsum) tasks. We apply SemSA on state-of-the-art large language models, including OPT-6.7B (Zhang et al.) and Llama2-7B (Touvron et al., 2023).

## 6.2 OVERALL PERFORMANCE

| Model | Method | Density (%) | WikiText-103 ppl ↓ | | WikiText2 ppl ↓ | | Attention Speedup ↑ |
|---|---|---|---|---|---|---|---|
| | | | w finetune | w/o finetune | w finetune | w/o finetune | |
| OPT-6.7B-2k | Casual (Zhang et al.) | 100 | - | 10.97 | - | 10.02 | 1.00 |
| | SpTrans (Child et al., 2019) | 30 | 12.87 | 46.78 | 12.20 | 38.03 | 3.48 |
| | Bigbird (Zaheer et al., 2020) | 33 | 11.38 | 12.77 | 10.81 | 11.74 | 4.01 |
| | Ours | **26** | **11.05** | **12.70** | **10.50** | **11.72** | **4.18** |
| Llama2-7B-4k | Casual (Touvron et al., 2023) | 100 | - | 8.39 | - | 23.84 | 1.00 |
| | SpTrans (Child et al., 2019) | 28 | 8.61 | 50.51 | 24.03 | overflow | 4.31 |
| | Bigbird (Zaheer et al., 2020) | 15 | **8.36** | 11.27 | 20.61 | 28.15 | 8.18 |
| | Ours | **10** | 8.40 | **10.89** | **20.51** | **27.49** | **11.67** |

Table 1: The perplexity and the attention speedup of different large language models with different sparse attention methods.

We evaluate different sparse attention methods in terms of both performance and wall-time speedup. For baseline methods, we choose Bigbird (Zaheer et al., 2020), the best BERT-based sparse attention method reported in Tay et al. (2020). We also include Sparse Transformer(SpTrans) designed for text generation tasks. As shown in Table 1, SemSA generates the sparse attention mask with the lowest density, while preserving good performance in terms of perplexity. Note that the official implementation of the Bigbird and Sparse Transformer algorithm hardly provides any speedup (Tay et al., 2020). Thus, we adopt the state-of-the-art framework, xFormers (Lefaudeux et al., 2022), to accelerate the sparse baseline methods. We can see that: (1) On OPT-6.7B with 2k tokens, SemSA obtains 26% density (4% ↓) of the attention map and 11.05 perplexity (0.33 ↑) after finetuning the model weights. (2) On Llama2-7B with 4k tokens, SemSA achieves 10% density (6% ↓) and 8.40 perplexity (1.99 ↑) after finetuning. With our kernel design, SemSA can achieve up to 4.18× and 11.67× attention speedup on OPT-6.7B-2k and Llama2-7B-4k, respectively. On wikitext-103 and wikitext2 datasets, we achieve lower perplexity at a lower density compared with spTrans and Bigbird, and demonstrate performance close to the original model on three subtasks of Longbench.

## 6.3 RUNTIME EFFICIENCY

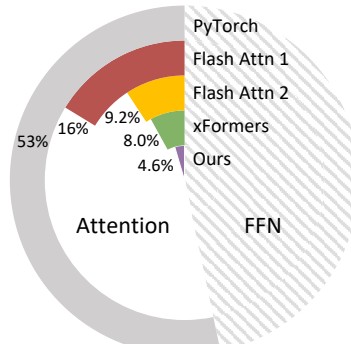

Figure 5: Runtime breakdown of Llama2-7B with 4k input length

| Model | Baseline | 2k token | 4k token |
|---|---|---|---|
| OPT-6.7B | PyTorch | 1.00 | 1.00 |
| | Flash Attn 1 | 1.10 | 2.00 |
| | Flash Attn 2 | 1.50 | 2.29 |
| | xFormers | 1.55 | 2.32 |
| | Ours | **1.58** | **2.34** |
| Llama2-7B | PyTorch | 1.00 | 1.00 |
| | Flash Attn 1 | 1.16 | 1.54 |
| | Flash Attn 2 | 1.25 | 1.72 |
| | xFormers | 1.27 | 1.81 |
| | Ours | **1.36** | **1.94** |

Table 3: End-to-end first-token-speedup of model with different attention implementations on A100.

| Method | Code Completion | Document QA | Few-shot Learning |
|---|---|---|---|
| Casual | 51.20 | 36.56 | 40.79 |
| SpTrans | 52.31 | **26.17** | 38.58 |
| Bigbird | 55.25 | 18.98 | 37.87 |
| Ours | **56.91** | 24.37 | **39.18** |

Table 2: The evaluation results on Longbench datasets of Llama2-7B-4k with different sparse attention methods.

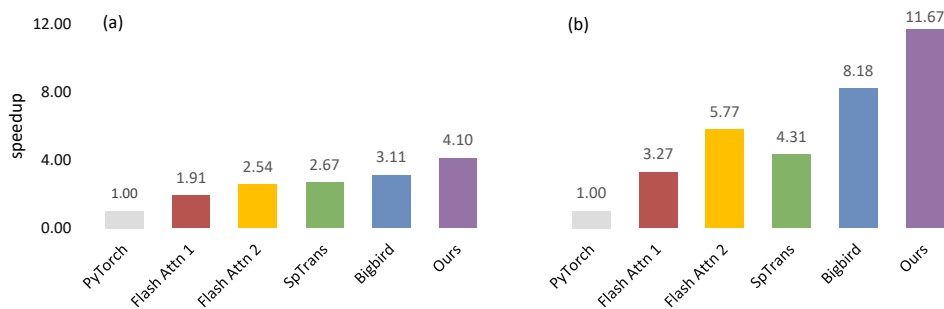

Figure 6: Attention speedup on Llama2-7B with (a) 2k tokens length and (b) 4k tokens length.

In this section, we reveal the efficiency of our attention kernel. Specifically, we adopt the sparse attention mask of SemSA, and implement it xFormer and our GPU kernels. We compare their runtime ratios and end-to-end speedups. We also add the Flash Attention for dense models. As shown in Figure 6, our kernels significantly reduce the runtime ratio to 4.6%, compared with the PyTorch (53%) and the state-of-the-art framework xFormers (8%). From Table 6.3, we can see that our kernels can consistently achieve higher end-to-end first-token-speedups on different models and with different token lengths.

## 6.4 ABLATION STUDY

We perform ablation study on Llama2-7B and use the perplexity results of WikiText-103 to validate the effectiveness of SemSA's components.

| Effect | Density(%) | Perplexity |
|---|---|---|
| w/o LP, w/o SC | 10.9 | 11.42 |
| w/o LP, w SC | 10.7 | 10.82 |
| w LP, w SC | 10.5 | **10.69** |

Table 4: Ablation study on loss propagation (LP) and softmax calibration (SC) for attention effect.

| Density Choice | Density(%) | Perplexity |
|---|---|---|
| Same for whole model | 10.7 | 11.03 |
| Same for each layer | 10.7 | 10.88 |
| Ours | 10.5 | **10.69** |

Table 5: Ablation study on mask generation and optimization. The effectiveness of using different masks for different heads.

### 6.4.1 EFFECTIVENESS OF EFFECT EVALUATION

SemSA's evaluation method in Theorem 1 considers the importance of each attention value through Loss Propagation (LP,e equation 2) and Softmax Calibration (SC, equation 3). As shown in Table 4, we evaluate LP and SC by directly substituting the attention effect $E$ of SemSA with different metrics: attention matrix $A$ (without LP, without SC) and $(\partial L/\partial A) \cdot A$ (without LP, with SC).

### 6.4.2 EFFECTIVENESS OF MASK GENERATION AND OPTIMIZATION

SemSA generates different masks with different density for different heads and layers. The influence of masks for the loss is minimized through by choosing the most suitable mask through optimization. We evaluate the effectiveness of such scheme by forcing SemSA to choose masks of the same density for each layer and for the whole model in Table 5.

## 7 CONCLUSION

In contrast to existing sparse attention methods that manually design uniform attention masks, our work proposes the semantic sparse attention (SemSA) method that decide non-uniform density and mask patterns for different attention heads. It accomplishes this by formulating and solving an optimization problem aimed at minimizing the loss change caused by sparsification. Compared with previous masks that manually define uniform attention mask pattern or density, SemSA preserves different semantics for different heads, yielding higher sparsity. Specifically, it achieves a $1.27\times$ to $1.6\times$ higher sparsity while maintaining superior accuracy. Furthermore, we implement GPU kernels to support the efficient computation of SemSA, and achieve a $1.36\times$ to $2.34\times$ end-to-end speedup for the first-token latency across multiple LLMs with varying context lengths.

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
