# OpenReview forum: "SemSA: Semantic Sparse Attention is hidden in Large Language Models."
_ICLR.cc/2024/Conference — ICLR 2024 Conference Withdrawn Submission_

### Official Review · Reviewer_CLpu · 2023-10-25

**Soundness:** 3 good
**Presentation:** 3 good
**Contribution:** 2 fair
**Rating:** 3
**Confidence:** 4

**Summary:**

The authors propose a new approach to sparse attention in large language models called SEMSA (Semantic Sparse Attention).

The authors argue that uniform sparse attention masks can lead to suboptimal performance, and instead propose optimizing different attention masks for different heads.

Empirically, the authors evaluate SEMSA on several benchmark datasets and show speedups with small accuracy loss, and that the learned masks capture meaningful semantic information. The authors show that it outperforms other sparse attention methods in terms of sparsity, speedup, and accuracy.

**Strengths:**

1. The paper is well written and the presentation of the idea is clear and straightforward to follow.

**Weaknesses:**

1. The author claims that they achieved speedup with negligible loss in accuracy, but table 1 says otherwise.

2. The speedup achieved by the proposed method is also somewhat unimpressive as SemSA can barely have any advantage over exact attention speedup methods like FlashAttention (as shown in Table 3).

3. Table 1 evaluation is only limited to Wiki dataset.

4. To argue for the superiority of SemSA, I believe the author should show the current landscape of acc/ppl vs efficiency tradeoff in sparsifying the attention and how SemSA breaks out of the current Pareto Frontier. Current experiments either show speedup or efficiency but do not systematically characterize the tradeoff.

**Questions:**

1. There is a tradeoff between ppl and efficiency. Figure 6 does not capture the tradeoff and is misleading in the sense that it's not clear the price that the authors paid to get such a speedup. Is it possible to juxtapose accuracy metrics with any efficiency metric in the efficiency evaluation?

2. "From Table 6.3, we can see that
our kernels can consistently achieve higher end-to-end first-token-speedups on different models and
with different token lengths."
Where is table 6.3?

---

### Official Review · Reviewer_1ev5 · 2023-10-29

**Soundness:** 3 good
**Presentation:** 3 good
**Contribution:** 2 fair
**Rating:** 5
**Confidence:** 4

**Summary:**

This work introduces a gradient-based method for quantifying the impact of masking out elements of the attention matrix in a trained Transformer. Additionally, the method generates attention pruning masks to improve the speed of the attention layer in the Transformer model. The effectiveness of the proposed method has been evaluated on 7B LLMs, resulting in attention wall-time speedup ranging from 4.18 to 11.67 compared to dense PyTorch models.

**Strengths:**

Formalizing the attention pruning task as an optimization problem is meaningful and generalizable, as it can be applied to different models and tasks.

The proposed method focuses on creating block-sparse masks rather than element-sparse masks, which offers advantages in terms of hardware compatibility and overall speedup.

**Weaknesses:**

My main concern regarding this work pertains to its scope of comparison. I think this work falls under the category of pruning methods rather than sparse attention construction, warranting a direct comparison with existing pruning methods. Gradient-based pruning methods [1] aim to estimate the importance of pruning units using gradient-based scores and then prune the least important ones. Equation 2 in this work can be derived from the commonly used form in gradient-based pruning methods. While the generation of block-sparse attention masks shares the same spirit with structural pruning, such as block pruning in [2], I am not saying that the novelty of this work is limited, but the scope could be discussing pruning techniques.

If we consider this work as a pruning study, comparisons with SpTrans and Bigbird may be insufficient. In addition, comparing the acceleration with Flash Attention and xFormer is also indirect, and it would be more relevant to compare it with other pruning methods.

[1] Accelerating Attention through Gradient-Based Learned Runtime Pruning
[2] Block Pruning For Faster Transformers

**Questions:**

Section 5 discussed the SRAM and said the implementation is based on Flash attention, so my understanding is that the implementation has already considered the IO of the attention layer. Is it correct?

---

### Official Review · Reviewer_xq6q · 2023-11-02

**Soundness:** 2 fair
**Presentation:** 2 fair
**Contribution:** 2 fair
**Rating:** 3
**Confidence:** 3

**Summary:**

This paper proposes a sparse attention method that calculates the importance of each attention element/block by estimating the loss change when masking that position. This method improves performance over previous baselines while maintaining speedup. This paper experiments on language modeling tasks and application tasks like code completion, and document qa.

**Strengths:**

1. The proposed method shows improvements over previous methods. If the proposal is indeed effective and open source, it will be used by many people.

**Weaknesses:**

1. Problematic theory: Attention effect evaluation is based on \textit{untenable} assumptions, and the calculation may be \textit{wrong}. First, only $\delta$ approaches 0, we can use first-order Taylor expansion, but the change in attention after being masked is not small enough to be close to 0, so we can not derive formulas 2 and 4. Second, Formula 3 seems wrong. For the first part,  is the condition "n=j"? For the second part, the influences of other attention are wrong, it should be +1 but not -1.  Therefore, Formula 5 which is a combination of Formula 3 and Formula 4 is questionable.

2. Problematic speedup: Since this method generates masks for each sequence and each layer, the method does not only need to calculate the full attention matrix but also needs to calculate gradients even during the forward pass.  So I doubt whether we can reduce latency using this method.

3. This paper is not easy to follow. For example,  what do you mean by "mean attention matrix" in Figure 5; Formula 6 is complicated but lacks descriptions; Section 4 is orthogonal to other parts, i.e., sparse attention method.  The title is more suitable for calling it a sparsization method based on the loss influence of attention masks.

4. This paper needs to perform more comparisons, refer to "Questions".

**Questions:**

1.  We would like to the the overall latency beyond attention speedup if we want to apply this technique.

2. "Explicit Sparse Transformer, Axiv 2019" has proposed a top-k sparse attention method, which is a simplified version of this paper. Therefore, this paper needs to compare it.

3. Does your performance drop actually come from the continual training on llama but your method?

4. Why not test on longer sequences?

5. Can you compare your findings on functional heads with studies in the age of Bert? Why is the surprise the semantic sparse attention is hidden in LLMs, is it hidden in all transformers?

---

### Official Review · Reviewer_xR4c · 2023-11-09

**Soundness:** 3 good
**Presentation:** 4 excellent
**Contribution:** 4 excellent
**Rating:** 5
**Confidence:** 3

**Summary:**

There are two mainstream approaches to achieve sparse attention. One approach is to manually design a uniform sparse attention mask for all attention heads. Another approach is to use sparse masks to skip some attention computations.

The paper proposed to learn a different sparse patterns/masks for different heads, by solving an optimization problem to minimize the loss change when switch to a sparse attention method. It achieves 1.27x to 1.6x higher sparsity while maintaining the accuracy.

**Strengths:**

The paper proposed a new way to employ sparse attention: using different sparse patterns for different heads. It optimizes the loss between using dense and sparse attentions to learn the sparse attention masks.

The paper adds a new approach and demonstrate good empirical results with state-of-the-art models. It also provides interesting analysis. For example, one experiment demonstrate how the different patterns for different layers/heads outperforms single attention pattern for the whole model.

**Weaknesses:**

The optimization to learn the sparse patterns required an already trained model. So the sparse attention cannot be applied at pre-training stage when a new model is trained from scratch.

**Questions:**

Questions:
  - How the density bound d_t is chosen? How much is the density bound in the experiments.